# Major and Trace Elements in Moldavian Orchard Soil and Fruits: Assessment of Anthropogenic Contamination

**DOI:** 10.3390/ijerph17197112

**Published:** 2020-09-28

**Authors:** Inga Zinicovscaia, Rodica Sturza, Octavian Duliu, Dmitrii Grozdov, Svetlana Gundorina, Aliona Ghendov-Mosanu, Gheorghe Duca

**Affiliations:** 1Joint Institute for Nuclear Research, Joliot-Curie Street 6, 1419890 Dubna, Russian; zinikovskaia@mail.ru (I.Z.); dsgrozdov@rambler.ru (D.G.); sgun@nf.jinr.ru (S.G.); 2Horia Hulubei National Institute for R&D in Physics and Nuclear Engineering, 30 Reactorului Street MG-6, 077125 Magurele, Romania; 3The Institute of Chemistry, 3, Academiei Street, 2028 Chisinau, Moldova; rodica.sturza@chim.utm.md (R.S.); duca@asm.md (G.D.); 4Department of Structure of Matter, Faculty of Physics, University of Bucharest, Earth and Atmospheric Physics and Astrophysics, 405, Atomistilor Street, 077125 Magurele, Romania; 5Faculty of Food Technology, Technical University of Moldova, 168, Stefan cel Mare Bv., 2004 Chisinau, Moldova; aliona.mosanu@tpa.utm.md

**Keywords:** fruit orchard, metal uptake by plants, potentially hazardous elements, environmental pollution

## Abstract

The correct assessment of the presence of potentially contaminating elements in soil, as well as in fruits cultivated and harvested from the same places has major importance for both the environment and human health. To address this task, in the case of the Republic of Moldova where the fruit production has a significant contribution to the gross domestic product, the mass fractions of 37 elements (Na, Mg, Al, Ca, Si, K, Mn, Fe, Sc, Ti, V, Cr, Co, Ni, Zn, As, Br, Rb, Sr, Zr, Mo, Cd, Sb, Cs, Ba, La, Ce, Nd, Sm, Eu, Tb, Yb, Hf, Ta, W, Th, and U) were determined by instrumental neutron activation analysis in soil collected from four Moldavian orchards. In the case of three types of fruits, grapes, apples, and plums, all of them collected from the same places, only 22 elements (Na, Mg, Cl, K, Ca, Sc, Mn, Fe, Co, Ni, Cu, Zn, As, Br, Rb, Sr, Sb, Cs, Ba, La, Th, and U) were detected. The enrichment factor, contamination factor, geo-accumulation index, as well as pollution load index were calculated to assess the soil contamination. At the same time, the metal uptake from the soil into fruits was estimated by means of transfer factors. Soil samples showed for almost all elements mass fractions closer to the upper continental crust with the exception of a slightly increased content of As, Br, and Sb, but without overpassing the officially defined alarm thresholds. In the case of fruits, the hazard quotients for all elements with the exception of Sb in fruits collected in two orchards were below unity. A subsequent discriminant analysis allowed grouping all fruits according to their type and provenance.

## 1. Introduction

The relationship between food and health becomes critically important as consumers now demand healthy, tasty, and natural products, grown in uncontaminated environments [1]. Consequently, the analysis of trace elements in fruits has gained considerable importance, as fruits, rich in carbohydrates, organic acids, as well as vitamins and minerals, are important components of human diet [2,3,4]. The potential beneficial health effects of fruits are also attributed to the phenolic compounds related to antioxidant activity [5]. According to [6], the consumption of fruits and vegetables is helpful to reduce the risk of cardiovascular diseases and even prevent cancer. For vegetarians and vegans, the intake of minerals and trace elements from fruits becomes particularly vital [7].

Typical factors affecting the mineral composition of fruits are soil composition, climate conditions (temperature and light intensity), and agricultural practices [8]. Contamination of fruits with potentially hazardous elements may occur due to extensive use of fertilizers and metal-based pesticides. Absorption from the airborne deposits on the aerial parts, as well as from soils through root systems are the main pathway for contaminants. The use of contaminated water in irrigation also represents an important source of excessive accumulation of potentially toxic elements in fruits [3,9].

Assessment of the fruits’ chemical composition is important from several points of view: (i) to ensure that the levels of potentially hazardous elements in fruits meet national and international standards; (ii) to permit their differentiation based on their regional origin [3,10,11]. Despite the significant nutritional importance of fruits, the number of studies devoted to their elemental composition, and especially concerning the presence of potentially toxic elements, is relatively few. In this regard, [12] presented the mass fractions of 12 essential and potentially hazardous elements in 98 commercially available fresh fruits in Poland. In [13], the presence of 13 elements including the potential contaminants Co, Cr, Mn, Ni, Cu, Zn, and Pb in three varieties of sour cherry and table grape cultivars was evidenced. As in previous cases, atomic absorption spectrometry was used to assess the levels of Cu, Zn, Cd, and Pb in various fruits sold in Egyptian markets [9].

Among the highest sensitivity and highest accuracy analytical methods, Instrumental Neutron Activation Analysis (INAA) has been successfully used due to its capability to determine the presence of up to 45 different elements simultaneously in a wide range of matrices, including fruits [7,14]. This is done without any previous preparation of the samples, such as acid digestion, which is likely to induce unwanted systematic errors [15,16].

According to the Köppen-Geiger classification [17], the moderately continental climate of the Republic of Moldova can be classified as Dfb with annual rainfall decreasing from 600 mm in the north to about 400 mm in the south. This characteristic, together with an almost ubiquitous presence of high quality chenozem soils, represents favorable conditions for an intensive agriculture and horticulture. For this reason, the Republic of Moldova has gained a good reputation as a supplier of high-quality wines, fruits, and vegetable products in southeastern Europe [18]. This performance is due in great measure to the chernozem, a remarkable type of soil due to its fertility and resilience, which covers almost all the Moldavian territory [19,20,21]. Here, due to centuries of cropping, a significant part of the humus, the most precious component of chernozem, was lost, which at present requires different organic and inorganic amendments to maintain its fertility.

About two thirds of the agricultural land in Moldova is cultivated by large farms holding more than 100 ha of land and specialized in cereal and technical crops, mainly oriented towards export markets. According to the National Bureau of Statistics of the Republic of Moldova, in the period from 2014–2019, the production of fruits increased from 497 to 840 × 10${}^{3}$ tones and of grapes from 594 to 657 × 10${}^{3}$ tones [22].

These achievements were possible due to an intensive use of fertilizers and pesticides, sometimes from uncertified sources, which could affect the quality of the soil, as well as of the crops, with negative consequences on human health. For this reason, the main aims of the present research are: (i) to determine, by INAA, the elemental composition of soils and fruits collected in four orchards in the Republic of Moldova and to assess the potential anthropogenic contamination, (ii) to determine the values of the transfer factor and hazard quotients for the investigated fruits, and (iii) to establish to what extent the elemental composition can be useful as a fingerprint to differentiate fruits by region and by type. The results thus achieved, as well as their analysis and discussion are the object of the present study.

## 2. Results

### 2.1. Soils

The multi-elemental capability of INAA and Epithermal Neutron Activation Analysis (ENAA) permits determining the mass fractions of 37 major and trace elements in 13 soil samples collected in four agricultural zones, i.e., Criuleni, Ialoveni, Cahul, and Purcari (Figure 1). The final results concerning the mass fraction of the eight major, rock-forming elements—Na, Mg, Si, Al, K, Ca, Mn, and Fe—as well as of the other 31 trace elements—Sc, Ti, V, Cr, Co, Ni, Zn, As, Br, Rb, Sr, Zr, Mo, Cd, Sb, Cs, Ba, La, Ce, Nd, Sm, Eu, Tb, Dy, Tm, Yb, Hf, Ta, W, Th, and U—are presented in Table A1 together with the corresponding data for the Upper Continental Crust (UCC) [23] and Moldavian Average Soil (MAS) [24], while a complete list of all experimental results can be found at Mendeley Data, http://dx.doi.org/10.17632/fmhtdcs5mf.1.

The Spearman ρ correlation coefficient matrix, as well as other statistical tests, such as Tukey’s Q, Mann–Whitney’s U, or the Kruskal–Wallis test for equal medians, show that, at *p* < 0.05 (Bonferroni correction), the distribution of the mass fractions of major elements that compose the investigated soils is closer to that of the UCC [23] (Figure 2, Table A1). In the case of trace elements, the potential pollutants As, Zr, Cd, Sb, and, especially, Br present mass fractions significantly higher than those of the UCC [23]. Regardless of these anomalies, all soil samples show similar patterns concerning the mass fraction distribution of all investigated elements. Moreover, we point out the similarity between the distribution of trace elements reported in [11,24,25] and our results, which could represent a confirmation of our measurements. This finding is also well illustrated in Figure 2, which reproduces the distribution of the mass fractions of the considered elements together with the corresponding Standard Deviations (SD). For a better interpretation, all mass fractions are normalized to those of the UCC [23].

### 2.2. Fruits

The INAA, as mentioned before, permits determining the mass fractions of 22 elements (Na, Mg, Cl, K, Ca, Sc, Mn, Fe, Co, Ni, Cu, Zn, As, Br, Rb, Sr, Sb, Cs, Ba, La, Th, and U) in the analyzed fruits. The results are summarized in Table A2. According to [26], the analyzed elements can be classified into three groups: (i) major elements: Na, Cl, K, Ca, and Mg; (ii) enzymatic elements playing an important role in biological processes: Co, Fe, Zn, and Se; (iii) trace elements with no biological functions, such as Sb, As, Rare Earth Elements (REE), the lanthanides: Th, U, etc. All these elements enter the human body by the daily consumption of food such as vegetables and fruits, as well as animal sub-products.

Major elements K, Ca, Mg, Na, and Cl present a relatively large domain of variation concerning mass fractions as the data reproduced in Table A2 confirm. Higher values are recorded for K in plums (42.8 ± 5.7 g kg${}^{-1}$), apples (41 ±1.9 g kg${}^{-1}$), and grapes (31.6 ± 3.3 g kg${}^{-1}$), followed by Ca, the mass fractions of which are about ten times lower, with maximum values of 5.9 ± 1.1 g kg${}^{-1}$ being observed for grapes. The mass fractions of the other three elements Mg, Na, and Cl are significantly lower. As a general remark, the greater variability of the mass fractions characterized by the Coefficient of Variation (*CV*), defined as the ratio of the standard deviation to the mean value [27], ranging between 10 for K and 101 for Na, makes any ranking difficult (Table A2).

The second group of elements includes Fe, Mn, Co, Cu, Zn, Ni, and Br, known to be either essential for humans due to their important biological roles [13] or as enzymes in plant metabolism, as is the case of Fe, Cu, and Zn [26]. According to Table A2, Fe presents the highest mass fraction in all analyzed fruits followed by Cu and Zn, whose maximum mass fractions are observed in grapes. In our opinion, this fact can be explained by the use of copper sulfate, which is mixed with calcium hydroxide to form the Bordeaux mixture used as a fungicide [13,28]. Generally, in plants, the mass fraction of Cu is inadequate for normal growth. However, the application of micronutrient fertilizers and copper-based fungicides may sometimes increase it alarming levels [9].

Manganese is also an essential element playing a cofactor role in several classes of enzymes [12]. In our case, its extremal mass fractions were reached for two sorts of grape from Criuleni 1.6 μg kg${}^{-1}$ and Cahul 8.6 μg kg${}^{-1}$, respectively, again asking for a greater variability of trace elements in investigated fruits.

According to [12], Ni and Co are important for hormonal activity, lipid metabolism, the activation of some enzymes, of the stabilization of DNA and RNA. In plums, Ni reaches extremal mass fraction values, which fluctuate between 0.7 and 1.6 μg kg${}^{-1}$, while the Br mass fractions are almost the same for all analyzed fruits.

The third group of elements consists of Sc, As, Rb, Sr, Rb, Cs, Ba, La, Th, and U and do not play an active role in plant metabolism, their presence being influenced by the soil and, to a lesser extent, by airborne material. In the case of As, with a mass fraction of about 0.16 μg kg${}^{-1}$, comparable to the reference plant [26] (Table A2), its presence in fruits cannot be considered harmful.

## 3. Discussion

### 3.1. Soils

The resemblance between the mass fractions of the analyzed elements in soils and the UCC [23] (Figure 2, Table A1) and confirmed by more statistical tests could be explained by the fact that all locations are distributed within an area of about 4000 km${}^{2}$ belonging to the same geological formation, i.e., the Moldavian Platform. However, the mass fractions of potentially harmful elements As, Br, Cd, and Sb are in some cases higher than that of the UCC [23], the highest difference being observed for Br by a factor of 5.7 to 8.2 (Figure 2, Table A1).

In the absence of some unanimously accepted criteria concerning the level of soil contamination with potentially harmful elements, we used more indices such as the Enrichment Factor (*EF*) [29], the Contamination Factor (*CF*) [30], the Geo-accumulation Index (*Igeo*) [31], as well as the more general Pollution Load Index (*PLI*) [32]. According to the definition, the *EF* [29] represents the normalized mass fraction of a considered element to the Sc mass fraction in the sample, all of them being renormalized to the ratio between the mass fractions of the same element and Sc in a pristine, uncontaminated neighboring soil. In the absence of such an environment in the case of Moldavian soil, we considered the UCC [23] as the best approximation for an uncontaminated environment.

On the contrary, in the case of *CF* [30], *Igeo* [31], and *PLI* [32], we considered as a reference the minimum alert values of the mass fractions as stated by the national regulations of Moldova [33,34], the Russian Federation [35], and Romania [36], which represents in our opinion a more conservative approach.

To assess the degree of soil contamination, we refer only to those elements defined as contaminants by at last one of the national regulations mentioned before, i.e., V, Cr, Mn, Co, Ni, Zn, As, Br, Mo, Cd, Sb, and Ba (Table A1 and Table A3).

The presence of As, Br, Cd, and Sb in soil in relatively high mass fractions with respect to the UCC is most probably related to human activity through the intensive use of fertilizers and pesticides. For a more complete analysis, we took into account, besides As, Br, Cd, and Sb, eight other potential pollutant elements, i.e., V, Cr, Mn, Ni, Co, Zn, Mo, and Ba, although their content was relatively close to that of the UCC (Table A1). Another remark concerns Ba, which appears only in Romanian regulations, the threshold of which (400 mg kg${}^{-1}$) is lower than the UCC mass fraction of 630 mg kg${}^{-1}$. In spite of this fact, we included it in the list of potentially harmful elements according to the most conservative model hypothesis.

Regarding the higher mass fraction of Br in soils, it should be pointed out that considerable amounts of Br are used in agriculture as pesticides, i.e., fungicides, herbicides, and insecticides, mainly as methyl bromide and ethylene dibromide. Moreover, according to [37,38], small quantities of Br can be found in K fertilizers. All these facts could explain Br presence in soils, although the industrial importance of this element is rather small. These considerations could also be valid for the other potentially harmful elements As, Cd, and Sb, as in the vicinity of the investigated orchards, there are no important industrial activities that could be considered responsible for their presence.

For this reason, the results concerning the presence of Potentially Hazardous Elements (PHE) in the soil of four analyzed orchards appear somehow contradictory. If we consider the most restrictive regulations, only the As mass fraction overpasses the 2 mg kg${}^{-1}$ alert threshold according to the Russian Federation regulations [35], while the V, Cr, and Ba average mass fractions appear slightly higher than the Romanian regulation alert limits [36]. In this way, except As, the investigated soils could be considered legally uncontaminated. As the corresponding *CF*s were calculated based on officially established minimal thresholds for PHE, their values displayed in Table A1 reflect in fact the official regulations. The same conclusion is sustained by the *Igeo* [32], the values of which, according to Table A3, varied between 0.49 ± 0.05 and 0.51 ± 0.04, significantly lower than one, the maximum value for an unpolluted soil. This last statement should be considered with care as according to the definition, the *Igeo* is calculated as a geometric mean of more *CF*s, so that the greater the number of elements with lower *CF*s is, the smaller the resulting *Igeo*.

According to Table A3, the *EF* values were less than unity for Co, Mo and Ba, between one and three for V, Cr, Mn, Ni, and Zn, and higher than three only in the case of Sb and Br. These values point towards a highly polluted environment only in the case of Br, as a *EF* < 1 signifies a pristine environment, which for 1 < *EF* < 3, becomes moderately contaminated and, finally, severely polluted if the *EF* is greater than five [39].

If the official regulations are taken into account, the soil of all four Moldavian orchards could be considered almost uncontaminated, a hypothesis not sustained by the corresponding values of the *EF*. In our opinion, this discrepancy could be explained by the absence of a set of unanimously accepted numerical criteria to assess the contamination degree of soils.

### 3.2. Fruits

To quantify the soil-to-plant transfer of the analyzed elements, the *TF* appeared to be the most appropriate descriptor for each type of fruit (Figure 3) [40]. Again, the highest values we found were for K as this element presented higher mass fractions in all fruits. In an ad hoc classification based on the *TF* values, Rb was in second place, although its role in plant metabolism is still insufficiently elucidated. A possible explanation of this finding could be related to the fact that both K and Rb are alkaline elements whose atomic radii are relatively close: 243 and 265 pm (10${}^{-12}$ m), respectively. A similar situation was observed for Ca and Sr, the last one presenting a *TF* even higher than that of Ca. In this regard, it should be mentioned that As has an almost negligible *TF* which, together with the absolute values of the mass fractions, suggests an unimportant contamination.

This conclusion is also sustained by comparing the experimental values of the mass fractions (fresh weight) in the considered fruits with those of the World Health Organization [41], especially concerning the more harmful As and Sb (Table A4).

The final stage of this study consisted of estimating both the Daily Intake of Metal (*DIM*) [42] and the Hazard Quotient (*HQ*) [43] for the analyzed fruits. According to the data reproduced in Table A4, *DIM* [42] showed a great range of values, varying from element to element. The uptake of Co from the analyzed fruits was very low. The lowest uptake of Fe was found in fruits collected in the Criuleni region followed by Ialoveni, Purcari, and Cahul. The lowest accumulation of Fe was from plums, while the Mn mass fractions changed in the following order: grapes > apples > plums; while in the case of zinc, the order was grapes > plum > apple. The bioaccumulation of toxic elements As and Sb the from analyzed fruits was very low. The *HQ*values for all elements, except Sb, in fruits collected in the Criuleni and Cahul regions were below 1.0, suggesting that the analyzed fruits are safe for consumption.

### 3.3. Discriminant Analysis

To get more information concerning the similarities, as well as the dissimilarities among the investigated fruits, Discriminant Analysis (DA), as one of the most appropriate statistical methods of analysis, was used. Following this method, it was possible not only to discriminate between grapes, apples, and plums, but also to evidence the differences between grapes according to the vineyards where they were collected from.

The results of this analysis are better illustrated by the Root 2 vs. Root 1 biplot reproduced in Figure 4, as well as by the corresponding structure of Root 1 and Root 2. Given the reduced number of samples (seven varieties of grapes distributed over four vineyards and three varieties of apples and plums), the main contribution to DA was restrained to 10 elements (Na, Mg, Cl, K, Fe, Cu, Zn, As, and Rb) that showed the greatest variability, in order to assure the maximum discernibility between cases.

As can be observed in Figure 4, Root 1 showed a net separation between the apple and plum clusters, on the one hand, and grapes, on the other, while Root 2 showed a better discrimination between the apple and plum cluster and a partial overlap of the grape and apple ones. From this point of view, Root 1 and Root 2 showed a net difference between plums, apples, and grapes. Further, within the grape cluster, the Purcari samples formed a more homogeneous group, quite different with respect to those of Cahul, Criuleni, and Ialoveni.

By analyzing the structure of Root 1 and Root 2 reproduced in Figure 4 (inset), it can be remarked that while in the case of Root 1, only K, Ca, and Cu make a relatively significant contribution, in the case of Root 2, the contribution comes from more elements, i.e., Cl, Ca, Fe, Cu, Zn, As, and Rb. In view of this, it should be noted that, according to [26], Fe, Cu, and Zn belong to the group of enzymatic elements, while K and Ca represent some of the major constituents of vegetal tissue.

## 4. Materials and Methods

### 4.1. Sampling and Sample Preparation for Analysis

Soils samples were collected at depths varying between 10 and 20 cm to avoid topsoil contamination arising from the surrounding environment. A 10 cm diameter corer was used for this operation. In the studied area, as mentioned before, chernozem soils of a dark brownish greyish color predominated with pH values around 6.0. Soil samples were firstly air dried for 24 h, passed through a 2 mm stainless steel sieve, and finally, dried at 105 ${}^{{}^{\circ}}$C until constant weight.

Fruits were collected in September 2018 in four zones in the Republic of Moldova: southeast (Purcari), south (Cahul), center (Ialoveni), and Codru (Criuleni) (Figure 1). The following types of fruits were collected: in Purcari, the grapes Merlot, Feteasca Neagra, and Saperav; in Cahul, the grapes “Muscat de Hamburg”, and “Moldova”, apples, and plums; in Ialoveni, the grapes “Alb de Suruceni”, the apples “Golden”, and the plums “Vengherca”; in Criuleni, the grapes Moldova, apples, and plums.

The apple and plum orchards were fertilized with manure, irrigated with uncontaminated water, and cared for according to good agricultural practices. The same practices were used in the case of vineyards except irrigation, which was not used. The tree ages varied between 9 and 15 years, and in some cases greater. When collected, the apples and plums were ripenedwithin a proportion of 60–65 and 70–75 %, respectively, while the grapes were collected at full maturity. For a better statistic, for each type of fruit, about 1 kg of fresh material was collected from different trees and grapevines, washed several times with distilled water, and dried at 105 ${}^{{}^{\circ}}$C (convection drying) until constant weight. Then, samples were ashed inside a muffle furnace at 400 ${}^{{}^{\circ}}$C, a temperature lower than the sublimation or boiling point of potentially harmful elements As, Se, and Sb.

For the INAA, samples of about 0.1–0.2 g were packed in polyethylene bags for short-term and in aluminum cups for long-term irradiation, respectively.

### 4.2. Instrumental Neutron Activation Analysis

The elemental mass fractions of fruits and soil samples were determined by the INAA and ENAA at the IBR—2 Fast Pulsed Reactor of the Joint Institute of Nuclear Research (JINR), Dubna. The procedure for sample irradiation was described in detail in [25,44]. The mass fractions of the elements based on short-lived radionuclides Ca, Cl, V, Ti, Mg, Al, Si, and Mn were determined by irradiation, 1 min for soil and 3 min for fruits, at a thermal neutron fluency debit of 1.6 · 10${}^{13}$ cm${}^{-2}$ s${}^{-1}$. Irradiated samples were measured for 15 min. To determine the mass fraction of long half-life isotopes Na, Sc, Cr, Fe, Co, Ni, Zn, As, Se, Rb, Sr, Zr, Mo, Sb, Cs, Ba, La, Ce, Sm, Eu, Tb, Hf, Ta, W, Th, and U, a cadmium-screened irradiation channel for epithermal and fast neutrons at a fluency debit of 3.31 · 10${}^{12}$ cm${}^{-2}$ s${}^{-1}$ was used. The samples were irradiated for 3 days, repacked, and then, measured twice after for 4 and 20 days. The measurement time (or gamma spectrum recording) was 30 min and 1.5 h, respectively. The final gamma-ray spectra processing and determination of mass fractions for each considered element was performed using proprietary software developed at Frank Laboratory of Neutron Physics [45].

### 4.3. Quality Control

The quality control of the analytical measurements was assessed using certified reference materials: National Institute of Standards and Technology Standard Reference Material (SRM): SRM 575a—trace elements in tine needles (*Pinus taeda*), SRM 1573a—tomato leaves, SRM 1633c—trace elements in coal fly ash, SRM 2709—San Joaquin soil, and Joint Research Centre BCR 667—estuarine sediment. In these conditions, the maximum uncertainties were no greater than 10%. Final data were expressed as the mean ± one Standard Deviation (SD) of three replications for each analyzed sample.

### 4.4. Anthropogenic Contamination Indices

To assess the degree of anthropogenic influence on soil, there are a few descriptors that compare the mass fractions of possible contaminants with the mass fractions of the same elements in different reference media such as the UCC [23] or neighboring, uncontaminated soil. In the absence of any confident data concerning uncontaminated soil in the Republic of Moldova, we considered the UCC [23] as the reference and, as mentioned before, the minimum alert values of mass fractions as stated by national regulations. Each index has its advantages and drawbacks, so further, for a more comprehensive estimation, we considered, as mentioned before, the Enrichment Factor *EF* [29], the *CF* [30], the *Igeo* [31], as well as the *PLI* [32].

The *EF* for the element *i* is defined as:(1)$$EF_{i}=\frac{c_{i,s}\cdot c_{Sc,b}}{c_{Sc,i}\cdot c_{i,b}}$$
where $c_{i,s}$ is the mass fraction of PHE *i* in the soil sample and $c_{Sc,i}$ represents the Sc mass fraction in the same soil sample; $c_{i,b}$ and $c_{Sc,b}$ are the mass fractions of the same element and Sc, respectively, in a reference, uncontaminated material (in most situations, the UCC). Scandium was chosen as the reference element as its industrial use is almost negligible.

The anthropogenic enrichment of PHE in soil could also be described by the *CF*, defined as:(2)$$CF_{i}=\frac{c_{i}}{c_{b}}$$
where $c_{i}$ is the mass fraction of the considered element at any given site and $c_{b}$ represents the background level for the same element [46].

The *Igeo* index is closer to the *CF*, with some modifications:(3)$$Igeo_{i}=log_{2}\frac{c_{i}}{1.5\cdot c_{b}}$$

Here, the factor of 1.5 was introduced to minimize the effect of possible variations in the background [47].

In turn, the *PLI* represents the *n*th order geometric mean of an entire set of contamination factors $CF_{i}$ regarding the considered elements as follows:(4)$$PLI=\sqrt[{n}]{\prod_{i}^{n}CF_{i}}$$
where *n* represents the total number of potentially harmful elements.

### 4.5. Plant Transfer Factor

A similar approach was used to quantify the soil-to-plant transfer. This time, to assess metal accumulation from soil in different plant compartments, we used the *TF* [40], as defined by:(5)$$TF_{i}=\frac{c_{i,plant}}{c_{i,soil}}$$
where $c_{i,plant}$ represents the mass fraction of the *i*th element in the plant material and $c_{i,soil}$ is the mass fraction of the same element in the soil (both on a dry weight basis) where the plants were collected. Higher than unity *TF* values indicate a significant transfer from soil to plant, while a *TF* lower than unity indicates a poor response of plants towards absorption [40].

### 4.6. Risk Assessment

To assess the risk posed by some trace elements whose presence, in small amounts, is indispensable for human metabolism, but can be harmful for human health if their mass fractions overpass some thresholds, we used both the *DIM* [42] and *HQ* [43] indices. According to the WHO [41], we considered the elements Co, Fe, Mn, Ni, Zn, As, and Sb, the soil mass fractions of which, if they exceed some thresholds, could be considered as contaminants.

To estimate the *DIM* [42] and *HQ* [43], the mass fractions of the above-mentioned elements were recalculated from mg kg${}^{-1}$ dry weight to μg g${}^{-1}$ fresh weight. The calculation of the oral *DIM* from the soil from the place of cultivation through fruits was done using the following formula:(6)$$DIM_{i}=DFC\cdot MFS_{i}$$
where the Daily Fruit Consumption (*DFC*) is assumed to be 300 g per person [48], while *MFS* represents the average mass fraction of a considered element *i*, expressed in mg day${}^{-1}$ fresh weight.

In turn, the *HQ* [43] for element *i* was calculated by the following equation:(7)$$HQ_{i}=\frac{DIM_{i}}{ORD_{i}}$$
where the oral reference dose ($ORD_{i}$) [43] for the element *i* is expressed in mg kg${}^{-1}$ assuming a 70 kg body weight.

It is worth mentioning that a *HQ* [43] index under unity is considered as safe [49].

### 4.7. Statistical Data Analysis

To evidence any correlation between different varieties of fruits, DA was used. Accordingly, the fruits were classified and grouped as a function of the mass fractions of those elements that presented the greatest variance. In this case, DA was used according to the a priori definition of sample groups, i.e., grapes, apples, and plums. In this way, with the constraint introduced by the a priori characterization by types, it was possible to establish a better discrimination between types of fruits and, in the case of grapes, by taking into account their geographical provenance.

To perform this task, both Statsoft^®^ Statistica™10 and PAST 4.0 [50] software were used.

## 5. Conclusions

To assess the quality of orchard soil and the corresponding harvested fruits, the instrumental neutron activation analysis was used to determine the mass fraction of 37 major and trace elements in the soil and 22 elements in apples, grapes, and plums, all of them collected from four renowned agricultural zones of the Republic of Moldova.

The final data permitted calculating, in the case of soils, the contamination factor, the geo-accumulation index, as well as the pollution load index. Similar, in the case of fruits, the enrichment factor, as well as the daily intake of metal and the hazard quotient appeared to be the most representative for assessing the contamination degree.

A final analysis of showed that in the case of soil, the mass fractions of almost all investigated elements were close to the upper continental crust. This finding was also confirmed by the selected environmental pollution indices, which pointed towards an almost negligible soil contamination.

In the case of fruits, K proved to be the most abundant major element with respect to the enzymatic elements: Fe, Zn, and Cu. The transfer factor values for K and Rb were higher than 1.0, while for elements considered as environmental pollutants, lower than 1.0. Daily intake values calculated for Co, Fe, Mn, Ni, Zn, As, and Sb varied greatly depending on fruit type and place of provenance. The health quotients for all elements, except Sb in fruits collected from some locations, were lower than unity, which implies that all the analysed varieties of fruits are safe for human consumption.

A final discriminant analysis allowed classifying the analysed fruits by type and place of provenance, suggesting that even some small differences in the mass distribution of certain elements could be used to discriminate between different varieties of fruits.

In view of these results, the main conclusion of this study points towards an almost uncontaminated orchard soil, as well as the safe consumption of the harvested fruits.

## Figures and Tables

**Figure 1 ijerph-17-07112-f001:**
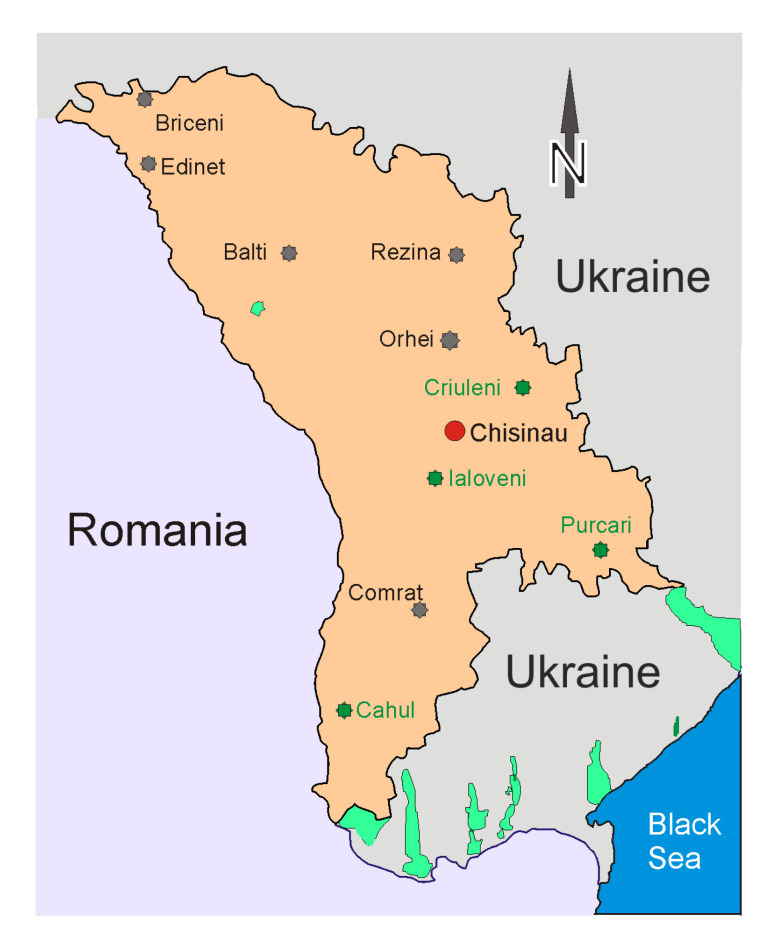
The geographical location of the sampling points (green stars).

**Figure 2 ijerph-17-07112-f002:**
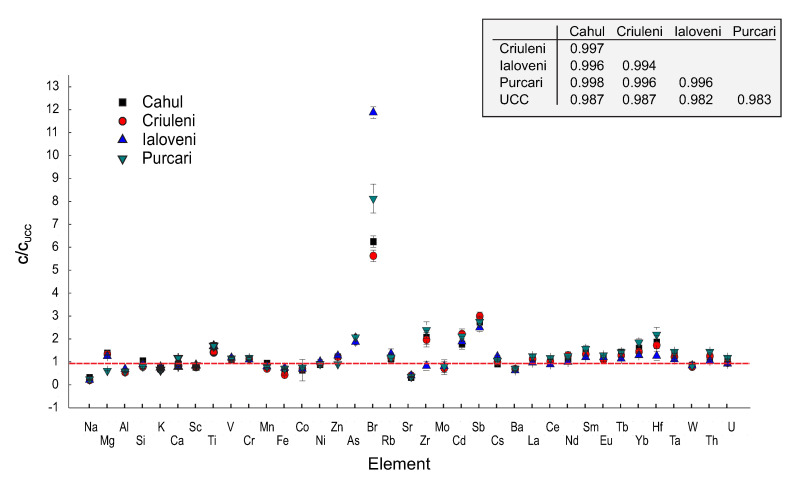
Mass fractions of major and trace elements (mass fractions ±1 SD) in soil samples normalized to the UCC [23]. The inset reproduces the Spearman’ ρ correlation coefficient matrix with Bonferroni correction at *p* < 0.01 calculated for all element except Br, Zr, Cd, and Sb.

**Figure 3 ijerph-17-07112-f003:**
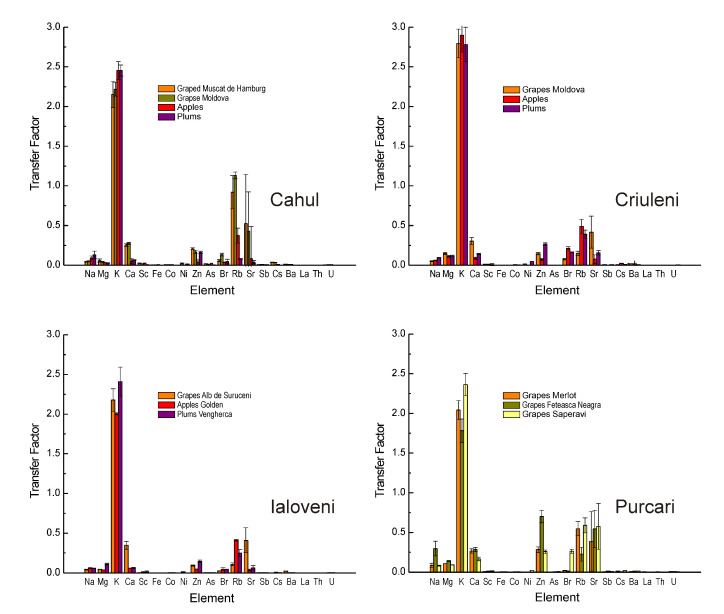
Transfer factor values in the system soil-fruit for fruits collected in four regions of the Republic of Moldova.

**Figure 4 ijerph-17-07112-f004:**
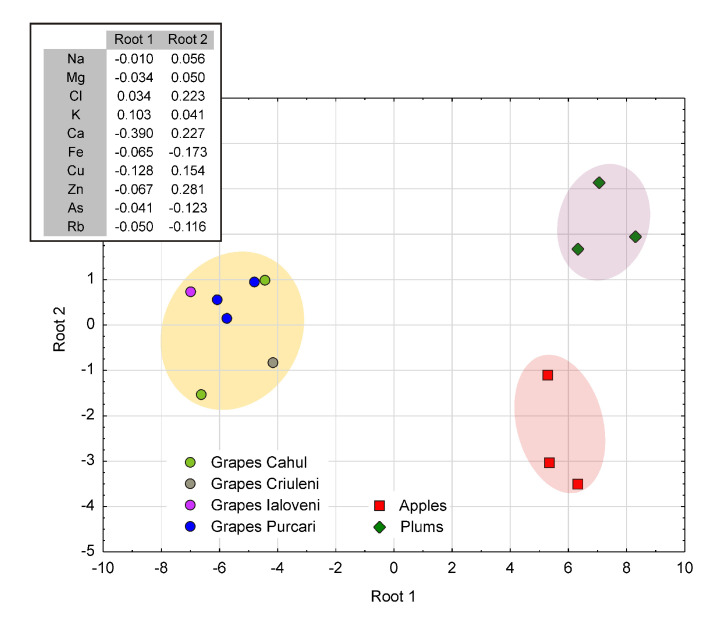
The result of discriminant analysis illustrating the existence of three clusters, each of them consisting of a single type of fruit.

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
