# Peer review of "Major and Trace Elements in Moldavian Orchard Soil and Fruits: Assessment of Anthropogenic Contamination"

_ijerph, 2020, doi:10.3390/ijerph17197112_

Round 1

Reviewer 1 Report

Submitted manuscript has an interesting and valuable merit concerning major and trace elements in Moldavian soils and fruits. Manusript is well structured, Materials and methods are clear and well described. Obtained results consist of soil and fruits subchapters, were 37 chemical elements were determined by Instrumental Neutron Activation in soil and  fruits in four regions – Cahul, Criuleni, Ialoveni and Purcari in Moldavia. The part of this chapter is the map of geographical location and graphs of mass fractions of major and trace elements and transfer factor values as well as result of Discriminant Analysis. Obtained results are valuable and well compared with ather authors. I appreciate way of evaluation  in soil – fruits system. Conclusions are clear and significant for next development of soil science and environment, as well.

Strengths of submitted manuscript is way of evaluation of chemical elements in soil and fruits using the Enrichment factor,  Contamination factor, Geo-accumulation Index and also Pollution Load Index. I appreciate also Discriminant analysis (DA) – appropriate statistic method of analysis which has been used in this work. The chapter of Results is relatively brief, chapter of Materials and Methods is often in order before chapter of Results, not before Conclusions as in this case, what could be weaknesses.

  • in Abstract 37 elements are listed, in Conclusions 39 elements in soil are mentioned
  • In the text is Table 1A, but in the headline of this table is Table A1 (not relevant)
  • I recommend better to use the units e.g. mg.kg-1, resp. g.kg-1 opposite mg/kg, resp. g/kg
  • I recommend to add a brief characterization of geological and soil conditions of 4 evaluated regions in Moldavian Platform
  • I recommend a slight revision of english language
    • on the page of 5, line 155 has to be:...consits of....
    • on the page of 15 (headline of Table A1) what does mean Pedological limits (PL) [?]
    on the page 19 in headline of Table A4. able 3 – what does it mean? Probably mistake

Author Response

Comments and Suggestions for Authors

Submitted manuscript has an interesting and valuable merit concerning major and trace elements in Moldavian soils and fruits. Manuscript is well structured, Materials and methods are clear and well described. Obtained results consist of soil and fruits subchapters, were 37 chemical elements were determined by Instrumental Neutron Activation in soil and fruits in four regions – Cahul, Criuleni, Ialoveni and Purcari in Moldavia. The part of this chapter is the map of geographical location and graphs of mass fractions of major and trace elements and transfer factor values as well as result of Discriminant Analysis. Obtained results are valuable and well compared with other authors. I appreciate way of evaluation in soil – fruits system. Conclusions are clear and significant for next development of soil science and environment, as well.

Strengths of submitted manuscript is way of evaluation of chemical elements in soil and fruits using the Enrichment factor, Contamination factor, Geo-accumulation Index and also Pollution Load Index. I appreciate also Discriminant analysis (DA) – appropriate statistic method of analysis which has been used in this work. The chapter of Results is relatively brief, chapter of Materials and Methods is often in order before chapter of Results, not before Conclusions as in this case, what could be weaknesses.

Answer: We have followed the Journal LATEX Template https://www.mdpi.com/data/MDPI_template.zip?v=20200817

which stated the following order: Abstract; Key words; Introduction; Results; Discussion; Materials and Methods; Conclusions; Acknowledgements; References; Appendices.

Remark: in Abstract 37 elements are listed, in Conclusions 39 elements in soil are mentioned

Answer: We have corrected

Remark: In the text is Table 1A, but in the headline of this table is Table A1 (not relevant)

Answer: Corrected everywhere appear Tables A1, A2, A3 and A4

Remark: I recommend better to use the units e.g. mg.kg-1, resp. g.kg-1 opposite mg/kg, resp. g/kg

Answer: Corrected everywhere

Remark: I recommend to add a brief characterization of geological and soil conditions of 4 evaluated regions in Moldavian Platform

Answer: In Introduction, we have added (Lines: 55-63) According to Köppen-Geiger classification [17], the moderately continental climate of Moldova can be classified as Dfb with annual rainfall decreasing from 600 mm in the north to about 400 mm in the south. This characteristic, together with an almost ubiquitous presence of high quality chenozem soil, represents favourable conditions for an intensive agriculture and horticulture. For this reason, the Republic of Moldova has gained a good reputation as a supplier of high-quality wine. fruit and vegetable products in south-eastern Europe [18]. This performance is due in great measure to the typical chernozem, a remarkable type of soil for its fertility and resilience, and which cover almost entire Moldavian territory [19–21]. Here, due to centuries of cropping, a significant part of humus was lost requiring different organic and inorganic amendments, to maintain its fertility.

Remark: I recommend a slight revision of English language

Answer: We have revised it carefully

Remark: on the page of 5, line 155 has to be:...consits of.…

Answer: Changed (Line 204) … consisting of ...

Remark: on the page of 15 (headline of Table A1) what does mean Pedological limits (PL) [?] on the page 19 in headline of Table A4. able 3 – what does it mean? Probably mistake

Answer: We have changed to (Lines: 88,89) Moldavian Average Soil (MAS) [24] and removed “abl3”

Reviewer 2 Report

MAJOR AND TRACE ELEMENTS IN MOLDAVIAN SOIL AND FRUITS: ASSESSMENT OF ANTHROPOGENIC CONTAMINATION

The manuscript investigates the anthropogenic potential contamination of orchard soils and fruits in 4 regions of Republic of Moldova for three kind of crops. This is an interesting and important topic to explore from a scientific point of view. However, the manuscript needs of several improvements and changes in order to be more readable and structured.

In my opinion, the MS cannot be accepted as it is for the publication in the journal but it needs of major revisions.

I’m writing below some general and specific comments, hoping that they could be useful for the authors.

Abstract

The abstract needs to be revised. I suggest to add a sentence at the beginning as introduction, in order to highlight the importance of the topic also linked to environmental and human health.

Keywords

I would suggest to remove a couple of keywords: fruits and discriminant analysis, replacing  them with more informative ones (e.g. metal uptake by plant, environmental pollution, fruit orchard…)

Title

Alternatively to the replacement of one keyword (fruit orchard), it could be also considered to be used in the title as follows:

Major and trace elements in Moldavian soil and fruits/fruit orchards: assessment of anthropogenic contamination

Introduction

The Introduction is well written, clear and precise in the description of the scenario.

Mat & met

My opinion is that it is better to add a first paragraph describing the fruit orchards where sampling has been done (crops, age of plants, agricultural practices applied, irrigation and fertilization management, water quality…). This information could help to better understand the results.

Results and Discussion

These sections need to be improved, in order to be more clear and linear for the reader. I wrote some specific points in the comments below.

Conclusions

The conclusion need to be rephrased trying to better describe the importance of the results of this study from scientific point of view in general and also for the Republic of Moldova.

Please check the text: in several points of the manuscript dot has to be replaced by comma

Line 37 -ii. to permit their differentiation based on their regional origin.

This point is not clear. Do you mean like a “fingerprint” of the product? Please re-write better.

Line 43 thR atomic absorption spectrometry

Line 46: INAA please add in full. Even if it is written in the abstract (where you should add the acronym), it should also be written the first time in full in the text.

Line 53: high quality soilS

Line 53-58: according to the first objective of the study (line 62), in this section should be added some info on the regions of Rep of Moldova.

Line 61: which could affect the quality of both soil and crops. I suggest to add something like “…affecting also environmental and human health”…

Line 66: I would suggest to replace “report” with “study”

LINE 70 ENAA please add in full

Figure 1: it should be better to move this figure in mat & met

Line 73: please remove K, is written twice

Line 77: http://dx.doi.org/10.17632/fmhtdcs5mf.1: the link doesn’t work

Line 82: UCC please add in full

Line 88: SD add in full and remove the full from line 279

Line 97: REE add in full

Figure 2. caption: Mass fractions of major and trace elements (mass fractions ± one SD) in soil samples normalized to the UCC [19] The inset reproduces the matrix of Spearman’s ρ correlation coefficients with Bonferroni correction at p < 0.01 calculated for all elements excepting Br, Zr, Cd and Sb.

Table A1. The mass fractions ± total experimental uncertainty of analysed soils elements. For comparison the corresponding values of UCC [19], Pedological Limits (PL) [? ] as well as National Reference Limits (NRL) for the Republic of Moldova [29,30], Russian Federation [31] and Romania [32] are reproduced too.

Tab A3 and A4, figures 3 and 4: my opinion is that it should be better to move this content in Results and refer each of them along the text.

Line 103 and 107: figure 2 not 1

Line 110-111: add in full the acronyms

Line 120-121 The presence of As, Br Cd and Sb in soil in relatively high mass fraction with respect to UCC is most probably related to human activity. Which kind of human activity are you referring? Is it possible due only to the agricultural inputs (water, fertilizers and pesticides), or there are other anthropogenic sources in the area?

Line 165 Coefficient of Variation [? ]: please add reference

Line 171 explained by the use of copper sulfate as fungicide. Did you verify that copper sulfate has been used in the sampled fruit orchards?

Line 193: Ru should be replaced by Rb

Line 214 add DA in the title

Line 240: a paragraph on the investigated sites/fruit orchards should be added (description of the fruit orchards, their location and management, see also comments on mat e met above). Considering also the information gave in the introduction on water and nutrition management and agricultural practices that could affect the mineral composition in fruits, all this information is necessary and important also for the discussion.  Line 248-255: it contains info that could be added in this new paragraph.

Line 242-255: how many fruits per sample? How many sampling times? Please be more precise in the description of the experimental trial and sampling.

Line 250-252: add that are different “varieties”

Line 282: to assess

Line 284 Rudnick and Gao, 2013: add number of ref

Line 309 (Mirecki et al., 2015): add number of ref

Line 318 DIM [39] and HQ [40] indices. Add in full

how to asses the correlation between the different type of fruits and the place where were cultivated,  region

Line 334-335 it is not clear how do you assessed the correlation between the different type of fruits and the place where were cultivated. Please, describe better.

Line 339 “a priori grouping of samples allowed to establish a better discrimination between types of fruits by taking into account their provenance”. Do you mean “geographical” provenance?

Author Response

Comments and Suggestions for Authors

MAJOR AND TRACE ELEMENTS IN MOLDAVIAN SOIL AND FRUITS: ASSESSMENT OF ANTHROPOGENIC CONTAMINATION

The manuscript investigates the anthropogenic potential contamination of orchard soils and fruits in 4 regions of Republic of Moldova for three kind of crops. This is an interesting and important topic to explore from a scientific point of view. However, the manuscript needs of several improvements and changes in order to be more readable and structured.

In my opinion, the MS cannot be accepted as it is for the publication in the journal but it needs of major revisions.

I’m writing below some general and specific comments, hoping that they could be useful for the authors.

Abstract

Remark: The abstract needs to be revised. I suggest to add a sentence at the beginning as introduction, in order to highlight the importance of the topic also linked to environmental and human health.

Answer: We have added at the Abstract beginning: A correct assessment of the presence of potentially contaminating elements in soil as well as in fruits cultivated and harvested from the same places has a major importance for both environmental and human health. To answer to this task, in the case of Republic of Moldova where the fruit production has a significant contribution to the Gross Domestic Product, ….

Keywords

Remark: I would suggest to remove a couple of keywords: fruits and discriminant analysis, replacing them with more informative ones (e.g. metal uptake by plant, environmental pollution, fruit orchard…)

Answer: We have changed to: fruit orchard; metal uptake by plant; potentially hazardous elements; environmental pollution

Title

Remark: Alternatively to the replacement of one keyword (fruit orchard), it could be also considered to be used in the title as follows: Major and trace elements in Moldavian soil and fruits/fruit orchards: assessment of anthropogenic contamination

Answer: We have changed to: Major and trace elements in Moldavian orchards soil and fruits: assessment of anthropogenic contamination

Introduction

The Introduction is well written, clear and precise in the description of the scenario.

Mat & met

Remark: My opinion is that it is better to add a first paragraph describing the fruit orchards where sampling has been done (crops, age of plants, agricultural practices applied, irrigation and fertilization management, water quality…). This information could help to better understand the results.

Answer: We have included (Lines 273 - 281) The apple and plums orchards were fertilized with manure, irrigated with uncontaminated water, and cared for according to good agricultural practice. The same practices were used in the case of vineyards excepting the irrigation which was not used. The tree ages varied between 9 and 15 years, and in some case greater. When collected, the apples and plums were ripped in a proportion of 60-65 and respectively 70 - 75 %, while the grapes were collected at full maturity. For a better statistic, for each sort of fruits, about 1 kg of fresh material was collected for from different trees and grapevines, washed several times with distilled water and dried at 105 o C (convection drying) until the constant weight.

Results and Discussion

These sections need to be improved, in order to be more clear and linear for the reader. I wrote some specific points in the comments below.

Conclusions

Remark: The conclusion need to be rephrased trying to better describe the importance of the results of this study from scientific point of view in general and also for the Republic of Moldova.

Answer: We have changed accordingly (Lines: 388-385) , including also the final sentence: “In view of these results, the main conclusion of this study points towards an almost uncontaminated orchards soil as well as to a safe consumption of harvested fruits.

Please check the text: in several points of the manuscript dot has to be replaced by comma

Remark: Line 37 -ii. to permit their differentiation based on their regional origin. This point is not clear. Do you mean like a “fingerprint” of the product? Please re- write better.

Answer: Rephrased (Line 75,77) … establish at which extent the elemental composition can be useful as a fingerprint to differentiate fruits by regions and by sorts. The results thus achieved as well as their analysis and discussion are the object of present study..

Remark: Line 43 thR atomic absorption spectrometry

Answer: We have removed thR

Remark: Line 46: INAA please add in full. Even if it is written in the abstract (where you should add the acronym), it should also be written the first time in full in the text.

Answer: We have changed here and wherever we have used for the firs time an acronym.

Remark: Line 53: high quality soilS

Answer: Corrected (Line 57) high quality chernozem soils

Remark: Line 53-58: according to the first objective of the study (line 62), in this section should be added some info on the regions of Rep of Moldova.

Answer: Completed (Line 55 - 63) We have added data concerning climate and chernozem soil.

Remark: Line 61: which could affect the quality of both soil and crops. I suggest to add something like “…affecting also environmental and human health”…

Answer: Corrected (Line 71 ) ….affect the quality of soil as well as crops with negative consequences on human health.

Remark: Line 66: I would suggest to replace “report” with “study”

Answer: We have replaced. Thank you for suggestion

Remark: LINE 70 ENAA please add in full

Answer: Done here, and everywhere the terms appeared for the first time.

Remark: Figure 1: it should be better to move this figure in mat & met

Answer: We have kept Fig. 1 in Introduction to explain to readers where  Rep. of Moldova is situated. As according to Journal template, Mat. & . Meth. succeed Discussion, we have considered more illustrative to present the Moldova’s map in Introduction.

Remark: Line 73: please remove K, is written twice

Answer: Removed

Remark: Line 77: http://dx.doi.org/10.17632/fmhtdcs5mf.1: the link doesn’t work

Answer: It is active now https://dx.doi.org/10.17632/fmhtdcs5mf.1

Remark: Line 82: UCC please add in full

Answer: Done

Remark: Line 88: SD add in full and remove the full from line 279

Answer: We have explained (Line 101) and used only acronym on the next document

Remark: Line 97: REE add in full

Answer: Done (line 101)

Remark: Figure 2. caption: Mass fractions of major and trace elements (mass fractions ± one SD) in soil samples normalized to the UCC [19] The inset reproduces the matrix of Spearman’s ρ correlation coefficients with Bonferroni correction at p < 0.01 calculated for all elements excepting Br, Zr, Cd and Sb.

Answer: Corrected (19 was removed)

Remark: Table A1. The mass fractions ± total experimental uncertainty of analysed soils elements. For comparison the corresponding values of UCC [19], Pedological Limits (PL) [? ] as well as National Reference Limits (NRL) for the Republic of Moldova [29,30], Russian Federation [31] and Romania [32] are reproduced too.

Answer: Corrected to: Table A1. The mass fractions ± total experimental uncertainty of analysed soils elements. For comparison the corresponding values of UCC [19], Moldavian Average Soil (MAS) [20] as well as National Reference Limits (NRL) for the Republic of Moldova [29,30], Russian Federation [31] and Romania [32] are reproduced too. Mass fractions expressed in mg kg−1 excepting major elements marked by * whose mass fractions are expressed in g kg−1. The elements considered as potentially hazardous according to [29–32] are marked with red colour. Total experimental uncertainty was calculated by composing the statistic error concerning the γ ray spectrum area for individual lines with reference material and neutron flux uncertainties.

NB we have replaced Pedological Limits by Moldavian Average Soil (MAS)

Remark: Tab A3 and A4, figures 3 and 4: my opinion is that it should be better to move this content in Results and refer each of them along the text.

Answer: We have moved only Figures because, in our opinion, these illustrate better the results. Moreover, Tables A3 and A4 are to ample for  the the journal page.

NB Although they are presented in Appendix, due to a unknown bug, are itemized as Table 3 and 4, instead of Table A3 and A4.

Remark: Line 103 and 107: figure 2 not 1

Answer: Corrected (Lines 84 and 268)

Remark: Line 110-111: add in full the acronyms

Answer: We have added everywhere

Remark: Line 120-121 The presence of As, Br Cd and Sb in soil in relatively high mass fraction with respect to UCC is most probably related to human activity. Which kind of human activity are you referring? Is it possible due only to the agricultural inputs (water, fertilizers and pesticides), or there are other anthropogenic sources in the area?Answer: We have added: (Lines: 147 – 149) This considerations could also be valid for the other potentially harmful elements As, Cd and Sb, as in the vicinity of investigated orchards there are no important industrial objectives which could be considered responsible for their presence.

Remark: Line 165 Coefficient of Variation [? ]: please add reference

Answer: We have explained{ Lines (182,183) ...fractions characterized by a Coefficient of Variation (CV) defined as the ratio of standard deviation to the mean value [38] ranging between 10 for K and 101 for Na, makes difficult any ranking (Table A2).

Remark: Line 171 explained by the use of copper sulfate as fungicide. Did you verify that copper sulfate has been used in the sampled fruit orchards?

Answer: We have include (Lines 188-191…) In our opinion, this fact can be explained by the use of copper sulfate which mixed with calcium hydroxide form the Bordeaux mixture used as fungicide [13,38]. Generally, in plants, the mass fraction of Cu is inadequate for a

Remark: Line 193: Ru should be replaced by Rb

Answer: Replaced

Remark: Line 214 add DA in the title

Answer: We have included together with the explanation of all acronyms.

Remark: Line 240: a paragraph on the investigated sites/fruit orchards should be added (description of the fruit orchards, their location and management, see also comments on mat e met above). Considering also the information gave in the introduction on water and nutrition management and agricultural practices that could affect the mineral composition in fruits, all this information is necessary and important also for the discussion. Line 248-255: it contains info that could be added in this new paragraph.

Answer: We have added (Lines 273 - 278) The apple and plums orchards were fertilized with manure, irrigated with uncontaminated water, and cared for according to good agricultural practice. The same practices were used in the case of vineyards excepting the irrigation which was not used. The tree ages varied between 9 and 15 years, and in some case greater. When collected, the apples and plums were ripped in a proportion of 60-65 and respectively 70 - 75 %, while the grapes were collected at full maturity. For a better statistic, for each sort of fruits, about 1 kg of fresh material was collected for from different trees and grapevines, washed several times with distilled water and dried at 105 o C (convection drying) until the constant weight.

Remark: Line 242-255: how many fruits per sample? How many sampling times? Please be more precise in the description of the experimental trial and sampling.

Answer: (Lines 273 - 278) When collected, the apples and plums were ripped in a proportion of 60-65 and respectively 70 - 75 %, while the grapes were collected at full maturity. For a better statistic, for each sort of fruits, about 1 kg of fresh material was collected for from different trees and grapevines, washed several times with distilled water and dried at 105 o C (convection drying) until the constant…….

Remark: Line 250-252: add that are different “varieties”

Answer: We have changed accordingly

Remark: Line 282: to assess

Answer: We have corrected

Remark: Line 284 Rudnick and Gao, 2013: add number of ref

Answer: Corrected The number is now [23]

Remark: Line 309 (Mirecki et al., 2015): add number of ref

Answer: Corrected the number is now [40]

Remark: Line 318 DIM [39] and HQ [40] indices. Add in full.

Answer: Added as the case of all acronyms

Remark: Line 334-335 it is not clear how do you assessed the correlation between the different type of fruits and the place where were cultivated. Please, describe better

Answer: We have used for this purpose the Discriminant Analysis, but taking into account the reduced number of places and fruits either, we have restrained to Discriminant Analysis. For the Principal Component Analysis, the number of samples (cases) was lower then the number of variables (elements), so that the best results were obtained by Discriminant Analysis.

Remark: Line 339 “a priori grouping of samples allowed to establish a better discrimination between types of fruits by taking into account their provenance”. Do you mean “geographical” provenance?

Answer: We have corrected (Line 362) … geographical provenance. Thank you

Reviewer 3 Report

This paper measures the amount of 37 elements, or a subset of them, in soil and in apples, grapes, and plums, and applies a variety of measures, pollution limits, and statistical techniques.  There is a lot of Bromine, and there is some excess Arsenic, but not in the fruit, Antimony, and Cadmium.  The conclusion is that the Bromine is polluting Moldovan soil and fruit, and while humans seem to accept Arsenic as a policy everywhere, it would be nice to reduce that.  The conclusion seems too positive, as pollution is discovered to be present, but the conclusion is vague and implies "no problem".  Promoting a clear set of standards is a good idea but not the topic of this paper.

Discriminant analysis shows that apples, grapes, and plums are different, which is not exciting, but the analysis is done correctly.

The writing of this paper appears to be unfinished.  There are three places where "?" seems to show that something was supposed to be checked or edited.  The coefficients of variation of Sodium and Potassium are 130 and 690 in the text (line 165) but 101 and 10 in Table A2.  The confusion about Potassium does not inspire confidence in the report.

All of the results are presented first, then all of the methods are presented.  That is an unusual order for a research paper.

Author Response

Comments and Suggestions for Authors

This paper measures the amount of 37 elements, or a subset of them, in soil and in apples, grapes, and plums, and applies a variety of measures, pollution limits, and statistical techniques. There is a lot of Bromine, and there is some excess Arsenic, but not in the fruit, Antimony, and Cadmium. The conclusion is that the Bromine is polluting Moldovan soil and fruit, and while humans seem to accept Arsenic as a policy everywhere, it would be nice to reduce that. The conclusion seems too positive, as pollution is discovered to be present, but the conclusion is vague and implies "no problem".

Remark: Promoting a clear set of standards is a good idea but not the topic of this paper.

Answer: We have reformulated (Lines 172-173) In our opinion, this discrepancy could be explained by the absence of a set of unanimously accepted numerical criteria to assess the contamination degree of soils

The discriminant analysis shows that apples, grapes, and plums are different, which is not exciting, but the analysis is done correctly.

Remark: The writing of this paper appears to be unfinished. There are three places where "?" seems to show that something was supposed to be checked or edited.

Answer: [?] appeared due to an incomplete link to the reference in the Reference list. We have included everywhere the correct citation according to LATEX template.

Remark: The coefficients of variation of Sodium and Potassium are 130 and 690 in the text (line 165) but 101 and 10 in Table A2. The confusion about Potassium does not inspire confidence in the report.

Answer: We have corrected it. (Lines 183) It was our mistake. Mea culpa.

Remark: All of the results are presented first, then all of the methods are presented. That is an unusual order for a research paper.

Answer: I totally agree, but I have followed the journal LATEX template: https://www.mdpi.com/data/MDPI_template.zip?v=20200817 which stated the following order: Abstract; Key words; Introduction; Results; Discussion; Materials and Methods; Conclusions; Acknowledgements; References; Appendices.

Reviewer 4 Report

In the paper entitled ‘’Major and Trace Elements in Moldavian Soil and Fruits: Assessment of Anthropogenic Contamination’’ the Authors major concerns are connected with elemental composition of soils and fruits, factors showing the level of  pollution and differentiation of fruits by four regions and type. The idea seems to be useful, but the manuscript has some flaws.

Firstly, the language should be improved in all aspects – grammar, vocabulary and sentence structure. Also, punctuation is the week side of the manuscript. Moreover, results part is insufficient and should be definitely better written.

More detailed comments are stated below.

Abstract

It should be strictly combined with the key conclusions. Please, rewrite.

Introduction

Lines 46-51 – Please, concise the paragraph. Now the second sentence is mostly the repetition of the first one.

Results

Results are insufficient and lack of more detailed presentation in a few more sentences/paragraphs. Results should be deeply rewritten to fulfill its function. It is not enough to write, e.g. ‘'The results are summarized in Table A3’’.

Why Fig. 1 is cited in line 82? The sentence (lines 79-82) has nothing in common with geographical localization.

What is the point in mentioning statistical tests in lines 79-80 instead of 4.7? Why Bonferroni was not mentioned?

Lines 94-97 The sentence should be removed to Materials and Methods. Classification does not belong to the achieved results. I also recommend mentioning Mn and Mo in the classification list.

Fig. 2 – Please, check if ‘’[19] 19’’ is correct.

Discussion

Discussion needs deeper insight into the consequences of the results. 

Lines 102-103 – Rethink the sentence; perhaps something like that is better ‘’The resemblance between mass fractions of analysed elements in soils and UCC (Figure 1, Table A1) could …”.

Lines 111-116 – Please, consider moving it to Materials and Methods.

Line 132 ’’rather reduced’’ or ’’small’’?

Line 150: ‘’EF 1 < EF < 3’’ – Please, improve.

Lines 160-163 – It is rather part of Results, but not Discussion. Why you have written ‘’As expected”? Please, explain the basis of your expectation.

Line 165 and Table A1 – What does [? ] mean?

Line 175 – Please, delete ‘’both animals and’’.

Lines 187-189 – Please, rewrite the sentence to be more clear.

Lines 206-210 – It belongs more to Results than to Discussion. Please, change it.

Lines 230-232 – Pucari – ‘’the best homogeneity’’ or ‘’quite different’’ – unclear.

Lines 234-236 – Again, it belongs more to Results than to Discussion. Please, change it.

Materials and Methods

In Materials and Methods 4.1. the lack of uncontaminated soil should be explained.

Why in Tab. A3 12 elements are mentioned? It should be explained in Materials and Methods.

Please, explain why in 4.2 not all elements were mentioned?

Line 282 – Perhaps ‘’soil, there were used …’’?

Line 313 – ‘’of from’’? Please, improve.

Line 323 – ‘’weight were in’’? Please, improve.

Line 325 – Please addthe literture after ‘’300 g per person’’.

Line 329 – Please explain what ‘’RfD’’ is.

Line 324 – Do you mean ‘’the place of cultivation’’?

Conclusions

This part should be rewritten to conclude the most important findings. In this part of the manuscript abbreviations should be used.

Lines 352-354 – Unclear sentence. Please, rewrite.

Lines 357-358 – What does it really mean that values are higher or lower?

Tables citation

The order of tables citation in the text should be changed according to the general rules. The soil results appear first (2.1), thus all the tables concerning soil should be cited here. Next, in the subchapter about fruits (2.2) all tables concerning fruits should be cited. Example, Tables A2 and A4 appear only in Discussion (lines 160 and 203, respectively), but they are not cited in the Results.

All abbreviations should be fully explained when first used, e.g.

Line 46 – INAA

Line 70 – ENAA

Line 75 – UCC

Line 205 – DIM, HQ

Line 272 – FLNP

Please, check the whole text of the manuscript.

Punctuation issues, for example:

Line 19: ‘’healthy. tasty and natural products. Grown‘’ – wrong full stops

Line 20 ‘’regard. The‘’ – wrong full stop

Line 23 ‘’According to [6]. The‘’ – wrong full stop

Line 24 ‘’For vegetarians and vegans. ‘’ – wrong full stop

Line 53 ‘’Republic of Moldova. Due‘’ – wrong full stop

Line54 ‘’wine. Fruit‘’ – wrong full stop

Lines 122, 236 ‘’i,e, ‘’ – wrong comas

Line 129 ‘’pesticides i.e.‘’ – lack of comma before ‘’i.e.’’

Line 176 – lack of comma after ‘’[12]

Line 270 – add comma before ‘’respectively’’

and many, many more throughout the text … Please, improve them carefully.

Doubled space bar, e.g.:

Line 28 ‘’[8].  Contamination‘’

Line 37 ‘’[3,10,11]. Despite‘’.

Lack of the space bar, e.g.

Lines 263, 286, 300.

Editorial mistakes, e.g.

Line 43 ‘’thr atomic‘’

Line 73 doubled ‘’K’’

Line 97 – what ‘’REE’’ mean?

Line 201 ‘’valued’’.

Language errors, e.g.

Line 42 ‘’potentially contaminants‘’

Line 60 ‘’an intensively use‘’

Line 79 ‘’statistic tests‘’

Line 128 ‘’it should pointed‘’

Line 148 ‘’values points‘’

Line 154 – the second part of the sentence

Line 159 ‘’a relative great domain’’

Lines 196, 197 – the ending of the sentences

Line 244 – should be ‘’predominated’’

and many more throught the text.

Why lines 476-478 are on p. 17 between tables?

Author Response

There were only three reviewers.

Round 2

Reviewer 2 Report

The manuscript has been implemented by the authors according to the suggestions. It can be considered for publication after minor revision of text editing, for example as follows:

Line 199 According to [12], Ni and Co play numerous roles in hormonal activity. lipid metabolism... 

Author Response

Review Report Form # 2

Open Review

(x) I would not like to sign my review report

( ) I would like to sign my review report

English language and style

( ) Extensive editing of English language and style required

( ) Moderate English changes required

( ) English language and style are fine/minor spell check required

(x) I don't feel qualified to judge about the English language and style

Yes Can be improved Must be improved Not applicable

Does the introduction provide sufficient background and include all relevant references?

(x) ( ) ( ) ( )

Is the research design appropriate?

( ) (x) ( ) ( )

Are the methods adequately described?

( ) (x) ( ) ( )

Are the results clearly presented?

( ) ( ) (x) ( )

Are the conclusions supported by the results?

( ) ( ) (x) ( )

Comments and Suggestions for Authors (Red Highlight)

MAJOR AND TRACE ELEMENTS IN MOLDAVIAN SOIL AND FRUITS: ASSESSMENT OF ANTHROPOGENIC CONTAMINATION

The manuscript investigates the anthropogenic potential contamination of orchard soils and fruits in 4 regions of Republic of Moldova for three kind of crops. This is an interesting and important topic to explore from a scientific point of view. However, the manuscript needs of several improvements and changes in order to be more readable and structured.

In my opinion, the MS cannot be accepted as it is for the publication in the journal but it needs of major revisions.

I’m writing below some general and specific comments, hoping that they could be useful for the authors.

Abstract

Remark: The abstract needs to be revised. I suggest to add a sentence at the beginning as introduction, in order to highlight the importance of the topic also linked to environmental and human health. Answer: We have added at the beginning: A correct assessment of the presence of potentially contaminating elements in soil as well as in fruits cultivated and harvested from the same places has a major importance for both environmental and human health. To answer to this task, in the case of Republic of Moldova where the fruit production has a significant contribution to the Gross Domestic Product,

Keywords

Remark: I would suggest to remove a couple of keywords: fruits and discriminant analysis, replacing them with more informative ones (e.g. metal uptake by plant, environmental pollution, fruit orchard…)

Answer: We have changed to: fruit orchard; metal uptake by plant; potentially hazardous elements; environmental pollution

Title

Remark: Alternatively to the replacement of one keyword (fruit orchard), it could be also considered to be used in the title as follows: Major and trace elements in Moldavian soil and fruits/fruit orchards: assessment of anthropogenic contamination

Answer: We have changed to: Major and trace elements in Moldavian orchards soil and fruits: assessment of anthropogenic contamination

Introduction

The Introduction is well written, clear and precise in the description of the scenario.

Mat & met

Remark: My opinion is that it is better to add a first paragraph describing the fruit orchards where sampling has been done (crops, age of plants, agricultural practices applied, irrigation and fertilization management, water quality…). This information could help to better understand the results.

Answer: We have included (Lines 276-284) The apple and plums orchards were fertilized with manure, irrigated with uncontaminated water, and cared for according to good agricultural practice. The same practices were used in the case of vineyards excepting the irrigation which was not used. The tree ages varied between 9 and 15 years, and in some case greater. When collected, the apples and plums were ripped in a proportion of 60-65 and respectively 70 - 75 %, while the grapes were collected at full maturity. For a better statistic, for each sort of fruits, about 1 kg of fresh material was collected for from different trees and grapevines, washed several times with distilled water and dried at 105 o C (convection drying) until the constant weight.

Results and Discussion

These sections need to be improved, in order to be more clear and linear for the reader. I wrote some specific points in the comments below.

Conclusions

Remark: The conclusion need to be rephrased trying to better describe the importance of the results of this study from scientific point of view in general and also for the Republic of Moldova.

Answer: We have changed accordingly, be including also the final sentence: In view of these results, the main conclusion of this study points towards an almost uncontaminated orchards soil as well as to a safe consumption of harvested fruits.

Please check the text: in several points of the manuscript dot has to be replaced by comma

Remark: Line 37 -ii. to permit their differentiation based on their regional origin. This point is not clear. Do you mean like a “fingerprint” of the product? Please re-write better.

Answer: Rephrased (Line 76,77) … to establish at which extent the elemental composition can be useful as a fingerprint to differentiate fruits by region and by types.

Remark: Line 43 thR atomic absorption spectrometry

Answer: We have removed thR

Remark: Line 46: INAA please add in full. Even if it is written in the abstract (where you should add the acronym), it should also be written the first time in full in the text.

Answer: We have changed here and wherever we have used for the firs time an acronym.

Remark: Line 53: high quality soilS

Answer: Corrected (Line 57) high quality chernozem soils

Remark: Line 53-58: according to the first objective of the study (line 62), in this section should be added some info on the regions of Rep of Moldova.

Answer: Completed (Line 54-61) We have added data concerning climate and chernozem soils

Remark: Line 61: which could affect the quality of both soil and crops. I suggest to add something like “…affecting also environmental and human health”…

Answer: Corrected (Line 71-72) ….affect the quality of soil as well as crops with a negative impact on human health.

Remark: Line 66: I would suggest to replace “report” with “study”

Answer: We have replaced. Thank you for suggestion

Remark: LINE 70 ENAA please add in full

Answer: Done everywhere it was necessary for the firs time to use an acronym

Remark: Figure 1: it should be better to move this figure in mat & met

Answer Figure 1 illustrates also the entire Moldavian territory, so in our opinion it was more illustrative to move it into Introduction.

Remark: Line 73: please remove K, is written twice

Answer: Removed

Remark: Line 77: http://dx.doi.org/10.17632/fmhtdcs5mf.1: the link doesn’t work

Answer: It is active now https://dx.doi.org/10.17632/fmhtdcs5mf.1

Remark: Line 82: UCC please add in full

Answer: Done (see above mention)

Remark: Line 88: SD add in full and remove the full from line 279

Answer: We have removed

Remark: Line 97: REE add in full

Answer: We have everywhere explained the acronyms

Remark: Figure 2. caption: Mass fractions of major and trace elements (mass fractions ± one SD) in soil samples normalized to the UCC [19] The inset reproduces the matrix of Spearman’s ρ correlation coefficients with Bonferroni correction at p < 0.01 calculated for all elements excepting Br, Zr, Cd and Sb.

Answer: Corrected (19 was removed)

Remark: Table A1. The mass fractions ± total experimental uncertainty of analysed soils elements. For comparison the corresponding values of UCC [19], Limits (PL) [? ] as well as National Reference Limits (NRL) for the Republic of Moldova [29,30], Russian Federation [31] and Romania [32] are reproduced too.

Answer: Corrected: Table A1. The mass fractions ± total experimental uncertainty of analysed soils elements. For comparison the corresponding values of UCC [23], Moldavian Average Soil (MAS) [24] as well as National Reference Limits (NRL) for the Republic of Moldova [33,34], Russian Federation [35] and Romania [36] are reproduced too. Mass fractions expressed in mg kg−1 excepting major elements marked by * whose mass fractions are expressed in g kg−1 . The elements considered as potential hazardous according to [33–36] are marked with red colour. Total experimental uncertainty was calculated by composing the statistic error concerning the γ ray spectrum area for individual lines with reference material and neutron flux uncertainties.

Remark: Table A3 and A4, figures 3 and 4: my opinion is that it should be better to move this content in Results and refer each of them along the text.

Answer: We have referred now in Results section (Lines: 94 and 113) Fig. 3 was referred in Discussion section as it illustrates the distribution of fruits Transfer Factors.

Remark: Line 103 and 107: figure 2 not 1

Answer: Corrected (Lines 94, 99)

Remark: Line 110-111: add in full the acronyms

Answer: We have added everywhere

Remark: Line 120-121 The presence of As, Br Cd and Sb in soil in relatively high mass fraction with respect to UCC is most probably related to human activity. Which kind of human activity are you referring? Is it possible due only to the agricultural inputs (water, fertilizers and pesticides), or there are other anthropogenic sources in the area?

Answer: We have explained: (Lines 171 -172) ...The presence of As, Br Cd and Sb in soil in relatively high mass fraction with respect to UCC is most probably related to human activity through the intensive use of fertilizers and pesticides. As well as (Lines: 184 ) These considerations could also be valid for the other potential harmful elements As, Cd and Sb, as in the vicinity of investigated orchards there are no important industrial objectives which could be considered responsible for their presence.

Remark: Line 165 Coefficient of Variation [? ]: please add reference

Answer: We have added:

Remark: Line 171 explained by the use of copper sulfate as fungicide. Did you verify that copper sulfate has been used in the sampled fruit orchards?

Answer: We have include (Lines124 - 128) In our opinion, this fact can be explained by the use of copper sulfate which mixed with calcium hydroxide form the Bordeaux mixture used as fungicide [13,38]. Generally, in plants, the mass fraction of Cu is inadequate for a

Remark: Line 193: Ru should be replaced by Rb

Answer: Replaced

Remark: Line 214 add DA in the title

Answer: We have included together with the explanation of all acronyms.

Remark: Line 240: a paragraph on the investigated sites/fruit orchards should be added (description of the fruit orchards, their location and management, see also comments on mat e met above). Considering also the information gave in the introduction on water and nutrition management and agricultural practices that could affect the mineral composition in fruits, all this information is necessary and important also for the discussion. Line 248-255: it contains info that could be added in this new paragraph.

Answer: We have added (Lines 122-123) ..as the case of Fe, Cu and Zn [26].

Remark: Line 242-255: how many fruits per sample? How many sampling times? Please be more precise in the description of the experimental trial and sampling.

Answer: We have added: (Lines 276 - 284) The apple and plums orchards were fertilized with manure, irrigated with uncontaminated water, and cared for according to good agricultural practice. The same practices were used in the case of vineyards excepting the irrigation which was not used. The tree ages varied between 9 and 15 years, and in some case greater. When collected, the apples and plums were ripped in a proportion of 60-65 and respectively 70 - 75 %, while the grapes were collected at full maturity. For a better statistic, for each sort of fruits, about 1 kg of fresh material was collected for from different trees and grapevines, washed several times with distilled water and dried at 105 o C (convection drying) until the constant…….

Remark: Line 250-252: add that are different “varieties”

Answer: We have changed accordingly

Remark: Line 282: to assess

Answer: We have corrected (Lines 210, 337)

Remark: Line 284 Rudnick and Gao, 2013: add number of ref

Answer: Corrected [19]

Remark: Line 309 (Mirecki et al., 2015): add number of ref

Answer: Corrected [37]

Remark: Line 318 DIM [39] and HQ [40] indices. Add in full.

Answer: Added as the case of all acronyms

Remark: Line 334-335 it is not clear how do you assessed the correlation between the different type of fruits and the place where were cultivated. Please, describe better.

Answer: We have changed to: (Lines 363-368) To evidence any correlation between different varieties of fruits, the DA was used. Accordingly, the fruits were are classified and grouped in function of the mass fraction of those elements which presented the greatest variance. In this case, the DA was used by a priori definition of sample groups, i.e. grapes grouped by vineyards, apples and plums. In this way, the constraint introduced by a priori characterization by types, it was possible to establish a better discrimination between types of fruits, and in the case of grapes, by taking into account their geographical provenance.

Remark: Line 339 “a priori grouping of samples allowed to establish a better discrimination between types of fruits by taking into account their provenance”. Do you mean “geographical” provenance?

Answer: We have corrected (Line 368) … geographical provenance. Thank you

Reviewer 4 Report

Dear Authors,

Most of the previously sent comments were not applied to the revised version of the manuscript and they remained unexplained. I am waiting for explantions concerning the previously sent comments.

Best regards.

Author Response

In the paper entitled ‘’Major and Trace Elements in Moldavian Soil and Fruits: Assessment of Anthropogenic Contamination’’ the Authors major concerns are connected with elemental composition of soils and fruits, factors showing the level of pollution and differentiation of fruits by four regions and type. The idea seems to be useful, but the manuscript has some flaws.

Firstly, the language should be improved in all aspects – grammar, vocabulary and sentence structure. Also, punctuation is the week side of the manuscript. Moreover, results part is insufficient and should be definitely better written.

More detailed comments are stated below. (light purple)

Abstract

Remark: It should be strictly combined with the key conclusions. Please, rewrite.

Answer: We have rewritten it, but we have taken int consideration the remarks of the other reviewers who suggested to include more data concerning the elemental composition of orchards soil and fruits as well the interpretation of our results within existing models.

Introduction

Remark: Lines 46-51 – Please, concise the paragraph. Now the second sentence is mostly the repetition of the first one.

Answer: Reformulated (Lines 48 - 52) Among the most high sensitivity, and high accuracy analytical methods, Instrumental Neutron Activation Analysis (INAA) has been successfully used due to its capability to determine the presence of up to 45 different elements simultaneously in a wide range of matrices. including fruits [7,14]. This is done without any previous preparation of the samples, such as acid digestion, likely to induce unwanted systematic errors [15,16].

Results

Remark: Results are insufficient and lack of more detailed presentation in a few more sentences/paragraphs. Results should be deeply rewritten to fulfil its function. It is not enough to write, e.g. ‘'The results are summarized in Table A3’’.

Answer: We have transferred more paragraphs from Discussion to Results, especially those concerning the investigated fruits.

Remark: Why Fig. 1 is cited in line 82? The sentence (lines 79-82) has nothing in common with geographical localization.

Answer: Corrected (Line 94) (Fig. 2, Table A1)

Remark: What is the point in mentioning statistical tests in lines 79-80 instead of 4.7 ? Why Bonferroni was not mentioned?

Answer: We have restrained to final results, otherwise we have to mention the value of Spearmans’ “rho” correlation coefficient and three more other parameters. We have mentioned the Bonferroni correction now.

Remark: Lines 94-97 The sentence should be removed to Materials and Methods. Classification does not belong to the achieved results. I also recommend mentioning Mn and Mo in the classification list.

Answer: Mo was not evidenced in fruits. Classification (Lines 108-110) was in our opinion necessary as in introduction to next paragraphs (Lines: 112-143)

Remark: Fig. 2 – Please, check if ‘’[19]19’’is correct

Answer: Now it is: UCC [23] and it is correct

Discussion

Discussion needs deeper insight into the consequences of the results.

Remark: Lines 102-103 – Rethink the sentence; perhaps something like that is better ‘’The resemblance between mass fractions of analysed elements in soils and UCC (Figure 1, Table A1) could …”.

Answer: We have removed (Line 188) ... results concerning the ...

Remark: Lines 111-116 – Please, consider moving it to Materials and Methods.

Answer: We have mentioned this paragraph in “Discussion” as it explain in a more coherent manner our approach in assessing the soil quality.

Remark; Line 132 ’’rather reduced’’ or ’’small’’?

Answer: We have changed to ..rather small. (Line 184)

Remark: Line 150: ‘’EF 1 < EF < 3’’ – Please, improve.

Answer: We have corrected (Line 204) “1 < EF < 3’’

Remark: Lines 160-163 – It is rather part of Results, but not Discussion. Why you have written ‘’As expected”? Please, explain the basis of your expectation..

Answer: We have moved to Results an entire section as it could represent a good transition from Results to Discussion (Lines 112 - 143)

Remark: Line 165 and Table A1 – What does [? ] mean?

Answer: We have corrected everywhere [?]. It appeared everywhere the LATEX has unrecognized a reference

Remark: Line 175 – Please, delete ‘’both animals and’’.

Answer: Deleted

Remark: Lines 187-189 – Please, rewrite the sentence to be more clear.

Answer: We have reformulated: (Lines 142,143) In the case of As, whose mass fraction of about 0.16 μg kg-1 is comparable with Reference Plant [26] (Table A2), its presence in fruits could not be considered as harmful.

Remark: Lines 206-210 – It belongs more to Results than to Discussion. Please, change it.

Answer: As mentioned before, we have moved to Results an entire section, including this one,

Remark: Lines 230-232 – Pucari – ‘’the best homogeneity’’ or ‘’quite different’’ – unclear.

Answer: We have changed to (Lines 252) Further, within the grape cluster, the Purcari samples formed a more homogeneous group, quite different with respect to Cahul, Ialoveni, and Purcari ones.

Remark: Lines 234-236 – Again, it belongs more to Results than to Discussion. Please, change it.

Answer: This paragraph (Lines 255-259) explains the results of Discriminant analysis, and, in our opinion, should remain here.

Materials and Methods

Remark: In Materials and Methods 4.1. the lack of uncontaminated soil should be explained.

Answer: First of all we have explained which media could be considered as a reference for uncontaminated soil which could be UCC or neighboring soil. In our case, we have mentioned that the UCC and minimum alert values of mass fractions as stated by national regulations were considered as references. Accordingly (Lines 313,314) In the absence of any confident date concerning uncontaminated soil in the Republic of Moldova, we have considered the UCC [23] as a reference, and, as mentioned before, the minimum alert values of mass fractions as stated by national regulations.

Remark: Why in Tab. A3 12 elements are mentioned? It should be explained in Materials and Methods.

Answer: We have explained (Lines 167 - 169) : To assess the degree of soil contamination, we referred only to those elements defined as contaminants by at last one of the national regulation mentioned before, I.e. V, Cr, Mn, Co, Ni, Zn, As, Br, Mo, Cd, Sb and Ba (Tables A1 and A3).

Remark: Please, explain why in 4.2 not all elements were mentioned?

Answer: In A2 we have mentioned all 22 elements whose content was experimentally determined by INAA. In the case of all other elements, their mass fractions were below the detection limits. For this reason, even in the Abstract we have mentioned only 22 elements in fruits.

Remark: Line 282 – Perhaps ‘’soil, there were used …’’?

Answer: We have changed (Lines 311 - 315) To assess the degree of anthropogenic influence on soil, there are few descriptors which compare the mass fraction of possible contaminants with the mass fraction of the same elements in different reference media such as UCC [23] or neighboring, uncontaminated soil.

Remark: Line 313 – ‘’of from’’? Please, improve.

Answer: Corrected (Line 338) we have removed “of”

Remark: Line 323 – ‘’weight were in’’? Please, improve.

Answer: Corrected (Line 340) we have removed “were”

Remark: Line 325 – Please add the literature after ‘’300 g per person’’.

Answer: We have added ref. [44]

Remark: Line 329 – Please explain what ‘’RfD’’ is.

Answer: Corrected (Line 357) We have replaced RfD by ORD (Oral Reference Dose) [43]

Remark: Line 324 – Do you mean ‘’the place of cultivation’’?

Answer: Corrected (Line …) “f .. place of provenance.. “

Conclusions

This part should be rewritten to conclude the most important findings. In this part of the manuscript, abbreviations should be used.

Remark: Lines 352-354 – Unclear sentence. Please, rewrite.

Answer: We removed this paragraph.

Remark: Lines 357-358 – What does it really mean that values are higher or lower? Tables citation

Answer: We have rephrased: (Lines 386 - 391) In the case of fruits, K proved to be the most abundant major elements with respect to enzymatic elements - Fe, Zn and Cu. The Transfer Factor values for K and Rb were higher than 1.0, while for elements considered as environmental pollutants lower than 1.0. Daily intake values calculated for Co, Fe, Mn, Ni, Zn, As, and Sb varies greatly depending on fruits sort and place of provenance. The Health Quotients for all elements, excepting Sb in fruits collected from some locations, were lower than unity which recommend all analysed varieties of fruits as safe for human consumption.

Remark: The order of tables citation in the text should be changed according to the general rules. The soil results appear first (2.1), thus all the tables concerning soil should be cited here. Next, in the subchapter about fruits (2.2) all tables concerning fruits should be cited.

Example, Tables A2 and A4 appear only in Discussion (lines 160 and 203, respectively), but they are not cited in the Results.

Answer: We have corrected

Remark: All abbreviations should be fully explained when first used, e.g. Line 46 – INAA, Line 70 – ENAA Line 75 – UCC, Line 205 – DIM, HQ Line 272 – FLNP

Answer: We have explained all acronyms where they were for the first time used

Remark: Please, check the whole text of the manuscript. Punctuation issues, for example: Line 19: ‘’healthy. tasty and natural products. Grown‘’ – wrong full stops. Line 20 ‘’regard. The‘’ – wrong full stop. Line 23 ‘’According to [6]. The‘’ – wrong full stop, Line 24 ‘’For vegetarians and vegans. ‘’ – wrong full stop Line 53 ‘’Republic of Moldova. Due‘’ – wrong full stop, Line54 ‘’wine. Fruit‘’ – wrong full stop, Lines 122, 236 ‘’i,e, ‘’ – wrong comas, Line 129 ‘’pesticides i.e.‘’ – lack of comma before ‘’i.e.’’ Line 176 – lack of a comma after ‘’[12], Line 270 – add comma before ‘’respectively’’, and many, many more throughout the text … Please, improve them carefully. Doubled space bar, e.g.: Line 28 ‘’[8]. Contamination‘’

Answer: We have thoroughly checked the entire text correcting as many as possible such kinds of misprints.

Remark: Line 37 ‘’[3,10,11]. Despite‘’. Lack of the space bar, e.g. Lines 263, 286, 300.

Answer: We have corrected all of them. No more double space or lack of space between words.

Remark: Editorial mistakes, e.g. Line 43 ‘’thr atomic‘’ Line 73 doubled ‘’K’’

Answer: Corrected both of them

Remark: Line 97 – what ‘’REE’’ mean? Line 201 ‘’valued’’.

Answer: We have explained (Line …) Rare Earth Elements - lanthanides

Language errors, e.g.

Remark: Line 42 ‘’potentially contaminants‘’ Line 60 ‘’an intensively use‘’

Answer: Corrected (Line.. et all other) potential (cca 25 places) and intensive (3 places)

Remark: Line 79 ‘’statistic tests‘’

Answer: We have included more “,”

Remark: Line 128 ‘’it should pointed‘’ Line 148 ‘’values points‘’

Answer: Corrected (Line 161)- should be pointed … (Line 203) ..values point ….

Remark: Line 154 – the second part of the sentence Line 159 ‘’a relative great domain’’

Answer: Corrected (Line 112) ... a relatively large domain ...

Remark: Lines 196, 197 – the ending of the sentences

Answer: Corrected

Remark: Line 244 – should be ‘’predominated’’ and many more throughout the text. Answer:

Answer: Corrected (Line 267) ...predominated...

Remark: Why lines 476-478 are on p. 17 between tables?

Answer: It is a bug of the LATEX template such as Table 3 and Table 4 instead of Table A3 and Table A4 (Lines …). These bugs will be corrected by the editor. It is not for the first time.

Round 3

Reviewer 4 Report

In my opinion, Authors have not corrected English. There are still editing flaws. Moreover, not all tables were cited in the results part.